# Upgrading Treatment and Molecular Diagnosis in Endometrial Cancer—Driving New Tools for Endometrial Preservation?

**DOI:** 10.3390/ijms24119780

**Published:** 2023-06-05

**Authors:** Miriam Dellino, Marco Cerbone, Antonio Simone Laganà, Amerigo Vitagliano, Antonella Vimercati, Marco Marinaccio, Giorgio Maria Baldini, Antonio Malvasi, Ettore Cicinelli, Gianluca Raffaello Damiani, Gerardo Cazzato, Eliano Cascardi

**Affiliations:** 1Obstetrics and Gynaecology Unit, Department of Biomedical Sciences and Human Oncology, University of Bari “Aldo Moro”, 70124 Bari, Italy; 2Unit of Gynecologic Oncology, ARNAS “Civico—Di Cristina—Benfratelli”, Department of Health Promotion, Mother and Child Care, Internal Medicine and Medical Specialties (PROMISE), University of Palermo, 90127 Palermo, Italy; 3MOMO’ FertiLIFE IVF Center, 76011 Bisceglie, Italy; 4Department of Emergency and Organ Transplantation, Pathology Section, University of Bari “Aldo Moro”, 70124 Bari, Italy; 5Department of Medical Sciences, University of Turin, 10124 Turin, Italy; 6Pathology Unit, FPO-IRCCS Candiolo Cancer Institute, 10060 Candiolo, Italy

**Keywords:** endometrial cancer, fertility preservation, molecular characterization, reproductive outcome

## Abstract

One emerging problem for onco-gynecologists is the incidence of premenopausal patients under 40 years of age diagnosed with stage I Endometrial Cancer (EC) who want to preserve their fertility. Our review aims to define a primary risk assessment that can help fertility experts and onco-gynecologists tailor personalized treatment and fertility-preserving strategies for fertile patients wishing to have children. We confirm that risk factors such as myometrial invasion and The International Federation of Gynecology and Obstetrics (FIGO) staging should be integrated into the novel molecular classification provided by The Cancer Genome Atlas (TCGA). We also corroborate the influence of classical risk factors such as obesity, Polycystic ovarian syndrome (PCOS), and diabetes mellitus to assess fertility outcomes. The fertility preservation options are inadequately discussed with women with a diagnosis of gynecological cancer. A multidisciplinary team of gynecologists, oncologists, and fertility specialists could increase patient satisfaction and improve fertility outcomes. The incidence and death rates of endometrial cancer are rising globally. International guidelines recommend radical hysterectomy and bilateral salpingo-oophorectomy as the standard of care for this cancer; however, fertility-sparing alternatives should be tailored to motivated women of reproductive age, establishing an appropriate cost–benefit balance between childbearing desire and cancer risk. New molecular classifications such as that of TCGA provide a robust supplementary risk assessment tool that can tailor the treatment options to the patient’s needs, curtail over- and under-treatment, and contribute to the spread of fertility-preserving strategies.

## 1. Epidemiology and Risk Factors

The term “gynecologic cancer” defines any cancer that arises in a woman’s reproductive system. During the period 2012–2016, 94,000 women were diagnosed with gynecologic cancer annually [1,2]. With an incidence of more than 3.6 million per year and mortality exceeding 1.3 million per year, these cancers constitute a public health issue, accounting for 40% of all cancer incidences and more than 30% of all cancer deaths in women worldwide [3,4]. Between 1990 and 2017, the age-standardized prevalence and incidence rate of EC increased globally by 0.89 percent and 0.58 percent a year, respectively, with a median age of diagnosis of 65 years [5,6,7]. According to recent statistics, incidence of EC is estimated to be 15% for patients ≤ 50 years old and 4–14% for those ≤40 years old [8]. For the year 2020, the World Health Organization estimated an EC incidence of 35,915 cases in women ≤ 44 years, 14,203 in women ≤ 39 years, and 2232 in women ≤ 29 years of age [9]. The Western lifestyle, with the associated spread of clustered risk factors such as excess weight, diabetes mellitus, hypertension, and high serum triglycerides, contributes to the emergence of EC [10]. PCOS, dysfunctional uterine bleeding/anovulation, and hypermenorrhea are other emerging risk factors linked to EC [9]. EC should be a trending topic for public health professionals: the IARC estimated that EC cases will increase by more than 50% worldwide by 2040 [5,11]. Different risk factors are associated with Type I and II EC, and partially mirror those of women’s risk for other cancers [2,12,13,14]. The main risk factor for type I EC is prolonged exposure to excess estrogen without adequate balance by progestin [15,16]. Sources of exogenous estrogen include hormone replacement therapy and Tamoxifen, while endogenous estrogen exposure may result from high body weight, dysfunctional menstrual cycles, or, in rare cases, tumors that secrete estrogen. Unopposed systemic estrogen therapy results in a high risk of endometrial hyperplasia (20–50% of women) [17] with a relative risk of EC ranging from 1.1 to 15 [18]. The use of the selective estrogen receptor modulator Tamoxifen increases the risk of EC in women after menopause with an effect that is duration- and dose-dependent with a RR of 3.32, and 95% CI 1.95–5.67 [19]. The dietary supplementation of phytoestrogens (non-steroidal chemical products with estrogenic and anti-estrogenic properties) for a time longer than 12 months may potentially increase the risk of EC (3.8%) [20]. The most common disorder associated with anovulation is polycystic ovary syndrome. Anovulatory women have an imbalance of sex hormones that leads to irregular uterine bleeding and continued proliferation of the endometrium that can lead to endometrial hyperplasia. Obese women have high endogenous estrogen levels and endocrine abnormalities such as altered levels of insulin-like growth factor and insulin resistance. Early menarche and late menopause increase the risk of the disease. Some ovarian cancers that produce estrogen, such as granulosa cell tumors, are most likely associated with endometrial neoplasia (25–50 per cent of women affected) and carcinoma (5–10 per cent of the affected) [21]. Some genetic syndromes are decisive risk factors for EC; Lynch syndrome, for example, accounts for 2–5% of all EC [22], and BRCA gene mutation significantly increases uterine cancer (RR 2.65, 95% CI 1.69–4.16) [23]. The risk of EC is significantly elevated, especially for BRCA mutation carriers taking the drug Tamoxifen [24]. Some risk factors that are associated with EC include nulliparity, infertility, hypertension, and diabetes [25]. Protective factors for type I EC include combined estrogen–progestin oral contraceptive use (decreases endometrial carcinoma risk by 30 per cent or higher [26]), childbearing at an older age, and breastfeeding. Cigarette smoking is associated with a diminished risk of EC in postmenopausal women relative risk (RR 0.71, 95% CI 0.65–0.78) [27]. Increased physical activity appears to reduce the risk of EC (RR 0.80, 95% CI 0.75–0.85) [28]. The habit of drinking coffee decreases the risk of EC with a dose-dependent rate. The reductions in risk for low/moderate drinkers were RR 0.87 (95% CI 0.78–0.97), and for heavy coffee drinkers were RR 0.64 (95% CI 0.48–0.86) [29]. Additionally, tea consumers have a decreased risk of EC proportional to the quantity consumed, especially for green tea (RR 0.8, 95% CI 0.7–0.9) [30]. Type II endometrial neoplasms have different risk factors than type I EC and are less well known because of their rarity. Obesity is less strongly correlated [31]. Pluriparity is a risk factor [32]. Type II tumors have a different racial distribution; type II EC is more common in Black women than in White women [33].

## 2. Classification and Molecular Aspects

Jan V. Bokhman, in a paper published in 1983, analyzed 336 patients affected by EC, and proposed a dualistic model based on pathogenetic and prognostic features. The model differentiates EC into two classes: Type I EC and Type II EC [34,35]. Type I EC represents more than 70% of cases and develops into a hyperestrogenism condition. These tumors are generally low-grade, estrogen-receptor-positive, endometrioid adenocarcinomas (80–90%) that arise from endometrial hyperplasia with atypia (atypical hyperplasia, AH/endometrial intraepithelial hyperplasia, EIN) and with positive prognostic outcomes. Type II tumors, accounting for 10% of EC, are clinically and histologically more aggressive and account for more than 40% of deaths from EC [10]. These neoplasms are predominantly non-estrogen-associated serous carcinomas, mainly receptor-negative, and usually arise in an atrophic endometrium from serous-type endometrial intraepithelial carcinoma. Type II tumors are more likely to be clear cell, papillary, serous, and undifferentiated carcinomas (10–20%), and carry a poor prognosis because of their high histological grade and high invasion and relapse rate [36]. TCGA project carried out a genomic analysis of 373 cancers of the endometrium, stratifying them into four distinct prognostic groups: polymerase and (POLE) ultra mutated, mismatch repair-deficient (MMRd), p53 mutant/abnormal (p53abn), and NSMP (non-specific molecular profile) [37,38]. The genes implied in the development of Type I and Type II EC are PTEN, mismatch repair proteins, β-catenin, KRAS and TP53-, E-cadherin, and PIK3CA, combined with additional molecular and pathologic biomarkers such as the expression of p53, L1CAM, estrogen receptor, progesterone receptor, and the presence of invasion of lymphovascular space (LVSI) to create an integrated risk profile. For example, the abnormal expression of the product of the gene TPp53 strongly correlates to the tumors’ high genomic instability and the consequent aggressive behavior of the tumor in progression and invasion. LVSI strongly correlates to lymphatic and capillary tumor spread. L1CAM expression is related to tumor cells with enhanced motility [39,40]. The new classification integrates molecular milieu with clinicopathological characteristics to define an accurate risk assessment [41].

## 3. The Risk Stratification Models

Type I endometrial cancer arises in cells exposed to high concentrations of estrogens. Many clinical conditions can lead to estrogen/progesterone imbalance, primarily high body weight (OR = 1.27; 95% CI, 1.17 to 1.38) and diabetes mellitus (OR = 1.20; 95% CI, 1.19 to 1.21; *p* < 0.0001). Obesity causes an increase in the risk of EC directly related to body mass index (BMI). For a BMI that ranges from 35 to 40, the risk of EC is increased 4-fold (OR = 4.45; 95% CI, 4.05 to 4.89; *p* < 0.0001), while for a BMI over 40 the risk of EC increases 7-fold (OR = 7.14; 95% CI 6.33 to 8.06; *p* < 0.0001). Other classically described risk factors include early menarche, failure to ovulate or infertility, PCOS, nulliparity, and late menopause [42]. Tamoxifen usage is associated with the insurgence of endometrial hyperplasia and dysplasia. A 10-year continuative therapy doubles the risk of EC (RR 2.29; *p* < 0.001) [43]. Protective factors are related to minor exposure to estrogens such as pluriparity, late menarche, and combined oral contraceptive and cigarette smoking [42]. Endometrioid EC type I arises based on atypical endometrial hyperplasia (AEH); concurrent AEH and EC is estimated in up to 29.5 percent of cases [43]. Autosomal dominant inherited syndromes such as Cowden, Lynch, and Peutz–Jeghers are associated with EC and account for approximately 2% to 5% of all cases. The most important is Lynch syndrome, which is caused by a deficit of the DNA mismatch repair proteins MSH.6, MLH.1, MSH.2, and PMS.2, and is associated with a lifetime risk of EC of 16–54% [44,45]. Cowden syndrome, caused by mutations of the protein PTEN tumor, has a lifetime risk for EC of up to 19–28% [45]. TCGA differentiates EC into four prognostically significant categories: POL-e (ultramutated), microsatellite instability (hyper-mutated), low copy number (CN-L) (endometrioid), and high copy number (serous-like) (CN-H) [46]. POLE is a DNA polymerase responsible for base excision repair. POLE-ultramutated tumors (4–12% of EC) have an extraordinarily high mutation rate and an excellent prognosis, regardless of tumor histotype and grade. Copy-number-high (MSI-H) tumors (23–36% of EC) are associated with PTEN, PIK3CA and PIK3R1 mutations [47], frequently with a low uterine location and Type I EC [47]. The TP53 gene encodes for the p53 protein, commonly mutated in cancer [48]. p53-abnormal tumors (8–24% of EC) have, typically, an aggressive behavior compared to other molecular subtypes; in fact, they are classically CN-H, serous, high-grade endometrioid/clear cell carcinomas and represent type II EC [48]. Low-grade endometrioid adenocarcinomas characterize CN-L tumors (30–60% of EC). These tumors are without a specific driver mutation, have no specific molecular profile (NSMP), and are also seen in type I EC. Regular expression of p53 tumors is common in endometrioid ER/PR-positive tumors, especially in obese patients [49,50]. TCGA subgroups are linked to the prognosis, progression-free survival, and risk profile of EC [51]. There is excellent prognosis in POLE-mutated tumors, and intermediate prognosis in MSI-H tumors and CN-L tumors, while CN-H tumors have poor outcomes [52,53]. The current joint guidelines of the European Societies of Gynaecological Oncology (ESGO), Radiotherapy and Oncology (ESTRO), and Pathology (ESP) proposed an integrated strategy which integrates the genome atlas molecular classification with EC characteristics such as myometrial invasion, histological type, and LVSI to define the correct treatment for these tumors [54]. This risk stratification model was summarized recently in 2022 by Crosbie et al. and is summarized in Table 1. [55]. The Pro-active Molecular Risk Classifier (ProMisE) proposed a novel molecular classification based on a combination of immunohistochemistry (IHC) for mismatch repair proteins, p53, and the molecular analysis of POLE [56]. The WHO proposed the algorithm shown in Figure 1 [55]. Raffone et al. calculated the accuracy of IHC for mismatch repair proteins in EC [57]. Assessing MMR status is crucial to propose humanized monoclonal antibody pembrolizumab and nivolumab to selected patients with a dramatic clinical improvement [58,59]. MMR-deficient tumors have increased resistance (2.1-fold) and recurrence (3.8-fold) compared to regular MMR expression. MMR specificity in recurrence rate of AEH/EC after initial regression is 100% [60].

The novel DNA analysis tools such as “liquid biopsy” and next-generation sequencing are already in use for the diagnosis of hereditary cancer syndromes but have not yet been routinely established in the clinical diagnostics of EC [61]. Multiple gene panels have been proposed for assessment: Bolivar et al., for example, proposed sequencing PIK3-CA PTEN, K-RAS, and CTTNB-1 to assess endometrioid EC [61]. Prognostic biomarkers have been proposed, including CTNNB-1 mutation status, ER/PR expression, and LVSI or L1 cell-adhesion molecule (L1CAM). LVSI is an independent prognostic marker that increases the mortality and recurrence and/or progression of the disease by 1.5–2-fold [62]. Overexpression of L1CAM is found in most aggressive ECs, especially in p53-abnormal tumors (80%). L1CAM is associated with aggressive behavior of the tumor (cell migration, invasion, epithelial–mesenchymal transition, and chemoresistance), and predicts worse outcomes (recurrence, reduced survival) in p53wilde-type/NSMP tumors [63]. Molecular classification of ECs and endometrial intraepithelial neoplasia (EIN) prior to conservative management is able to differentiate aggressive tumors which should be treated with primary surgery [64]. The rate of progression in patients with p53-abnormal tumors is 50%, whereas in POLE-mutated tumors it is just 25%, independent of the pathological findings [64]. An open field of study is the significance of the ProMisE classifier in fertility-preserving strategies [65].

## 4. Diagnostic Work Up

Abnormal uterine bleeding (AUB) and heavy menstrual bleeding (HMB) are common conditions affecting 19.5% of women of reproductive age [66]. In 2011, FIGO provided definitions of AUB, HMB, and Chronic AUB that should be used in medical literature and current clinical practice to standardize language [67]. The acronym PALM-COEIN could facilitate accurate diagnosis and treatment of uterine bleeding: PALM stands for the causes that can be assessed by imaging and pathology, such as polyps, adenomyosis, leiomyoma and malignancy, while the word COEIN stands for non-structural causes such as coagulation disease, ovulation problems, endometrial causes, iatrogenic, and others causes [68]. Abnormal bleeding is one of the main symptoms of all types of uterine disease, but has low specificity for malignancy. There is no correlation between AUB and the FIGO stage of the tumor; in addition, women not presenting AUB at the diagnosis of EC showed significantly better prognosis [69]. Bleeding disorders such as abnormal premenopausal and postmenopausal bleeding are the main EC symptoms for which patients seek gynecological consultation. Transvaginal ultrasonography is a safe, straightforward, and easy way to examine double-layered endometrial thickness and to triage women for further investigations [70]. According to international guidelines, gynecologists should assess endometrial thickness with transvaginal ultrasonography (TVUS) in women with abnormal bleeding that arises after menopause [71,72]. A thin endometrium should reassure clinicians about EC and lead to expectation management with seriated TVUS. In cases with a thickened endometrium, endometrial biopsy is warranted. TVUS diagnostic accuracy for EC diagnosis depends on the cut-off in use. The British Gynaecological Cancer Society guidelines currently recommend an endometrial thickness cut-off of ≥4 mm that has shown 94.8% sensitivity, 46.7% specificity, and a 99% negative predictive value for EC detection [70,71]. There is not consensus on the best cut-off to use to select AUB patients requiring endometrial biopsy; there is a high prevalence of EC in symptomatic patients when TVUS showed a thickness < 4 mm (8.5%). Some authors have suggested new diagnostic tools for the assessment of EC [73]. One of the main prognostic factors of EC is the depth of myometrial invasion [74], which strongly correlates with the 5-year survival rate: 94% for EC confined to the endometrium, 91% for EC in the inner 1/3 of the myometrium, and 59% when EC is in the outer 1/3 of the myometrium [75]. In addition, myometrial invasion correlates with the risk of extrauterine extension of EC: tumors confined to the inner 1/3 of the myometrium have a 12% risk of extrauterine extension while tumors invading the outer 1/3 have a 46% risk [76]. Contrast-enhanced magnetic resonance imaging (MRI), prevalently T1-weighted imaging including DCE MRI, is the most accurate diagnostic tool for the deep myometrium, with a sensitivity of 72–94% and a specificity of 87–96% [77,78,79,80]. Because of the high cost and technical issues related to MRI, physical examination followed by office vaginal ultrasound is a more affordable and accessible diagnostic technique proposed for deep myometrial invasion assessment, with an estimated sensitivity of 75% and specificity of 86% [81]. Biopsy provides the definitive diagnosis: hysteroscopy-guided biopsy remains the gold standard for diagnostic EC, (sensitivity 99.2%, specificity of 86.4%) [82]. Tao brush cytology and Pipelle have a positive predictive value of 81.7% and negative predictive value of 99.1%, but have sampling issues [83]. Blind dilatation and curettage has the highest undiagnosed rate for EC. Hysteroscopy with directed biopsy/curettage is more effective in diagnosing cervical involvement (specificity 98.71% vs. 93.76% (*p* < 0.01)) and more accurate in diagnosis of EC histology type and tumor grade than blind D and C. In Figure 2 we summarized the main ultrasonographic and hysteroscopic findings in endometrial hyperplasia and endometrial cancer.

## 5. Imaging: FIGO and TNM Staging

After the histological diagnosis, further investigations are performed to assess FIGO staging. A vaginal ultrasound may exclude concurrent cancers in the ovaries and presence of ascites, and can define myometrial invasion. A routine chest/abdomen/pelvis Computer Tomography (CT) scan should be performed on high-grade carcinomas to exclude metastatic disease. MRI should be used to differentiate EC from cervical cancer and assess soft tissue extension of EC [84]. MRI has shown low sensitivity (30.3%) for detecting metastatic lymph nodes; with PET/CT, this rate was 57.6% [85]. The low accuracy of imaging to assess lymph node involvement is the reason why accurate surgical staging remains the gold standard. A negative sentinel node evaluation confirms the pathologic absence of metastatic nodes (pN0) in patients with low/low–intermediate risk. Classical surgical lymph node staging remains a strategy of choice in patients with high–intermediate/high-risk disease [86]. Diagnostic laparoscopy is a minimally invasive surgical technique used for the assessment of intra-abdominal masses. It permits the direct inspection of intraabdominal organs and rules out endometrial cancer outside the myometrium or accompanying ovarian malignancies. Because of the relatively low incidence of synchronous endometrial and ovarian cancer (3–5%), a diagnostic laparoscopy is not mandatory in low-risk early EC, when there is no myometrium invasion, in grade 1 endometrial EC, in unsuspicious ovaries, and in normal cancer antigen 125 [87]. The final classification based on the operative staging of EC is that of the Tumour-Node-Metastasis (TNM)-Classification and FIGO, and is shown in Table 2 and Figure 1.

## 6. Conventional Treatment: NCCN Guidelines

The standard management of EC involves surgery, chemotherapy, and/or radiation therapy. The gold standard staging procedure for EC is total hysterectomy with bilateral sal-pingo-ovariectomy (TH/BSO) with, if necessary, lymph node surgical assessment [90]. In some selected premenopausal patients, ovary preservation may be a safe choice in stage I endometrioid cancer [91]. Minimally invasive surgery does not compromise oncological outcomes and has a lower rate of complications, so should be proposed in patients with macroscopically uterine-confined cancer. A LAP2 trial compared oncological outcomes in laparoscopic vs. laparotomic surgery, showing recurrence rates of 11.4% for LPS versus 10.2% for LPT surgery and a 5-year overall survival rate of up to 84.8% [92,93]. A trial by Maurits et al. [93] showed a significant complication rate of 14.6% in laparoscopy versus 14.9% in laparotomy, and a minor complication rate of 13.0% in laparoscopy versus 11.7% in laparotomy. Laparotomy remains the gold standard for patients with old age, a large uterus, or metastatic presentations [94]. Robotic surgery may be the surgical choice for the severely obese and for patients at higher anesthesiologic risk [95]. During the surgery, suspicious intraperitoneal areas and enlarged lymph nodes should be biopsied and peritoneal cytology should be collected. Through surgical staging, an accurate diagnosis, extension of the disease, a prognostic assessment and patients who require further adjuvant therapy can be defined. Routine lymph node dissection identifies patients with nodal localization requiring adjuvant treatment with radio and/or chemotherapy [96,97,98]. Guidelines recommend sentinel lymph node biopsy in patients with low-risk and intermediate-risk diseases. Radiotherapy plus brachytherapy, external beam radiation, and the combination of both, or chemotherapy with carboplatin with a given area under the free carboplatin plasma concentration versus time curve of 5–6 plus paclitaxel 175 mg/m^2^ are the standard adjuvant therapies that are proven to lower the risk of tumor recurrence. Adjuvant treatment recommendations for EC strongly depend on the prognostic risk group. For low-risk ECs, no adjuvant treatment is recommended [86]. In intermediate-risk populations, adjuvant brachytherapy should be proposed [86]. Adjuvant chemotherapy should be proposed in high-risk populations, especially for high grade and/or substantial LVSI. The omission of adjuvant treatment should be considered if a close follow-up is guaranteed. In stages III, IV, and recurrent EC, debulking surgery should be performed only if complete macroscopic resection is possible with acceptable morbidity. Primary chemotherapy should be used if debulking surgery is not feasible or acceptable [99].

### Immunotherapy as a New Approach in EC

One of the fields of interest in gynecological carcinomas is immunotherapy. The basic principle is that cancer grows when the host’s immune system is abnormal, and immunotherapy strengthens the patient’s immune system, allowing it to act better against cancer cells, slowing the growth and inhibiting the spread of the cancer [100]. Therefore, as in other female solid carcinomas [101], the evaluation of tumor infiltrating lymphocytes (TILs) has a fundamental role in predicting the response to immunotherapy. Currently, there are several immunotherapy strategies, among which those related to programmed cell death protein 1 (PD-1) and its ligand (PDL-1) are very encouraging. PD-1 and PDL-1 are proteins that inhibit the T-lymphocyte-mediated inflammatory response and allow cancer to evade apoptosis. PD-1 is a transmembrane protein expressed on the surface of lymphocytes which acts as an immunological checkpoint, i.e., it prevents the excessive activation of immune system cells from which immune and autoimmune responses arise [102]. Using anti-PD1 or PDL-1 molecules, immunotherapy inhibits the immune inhibitory system and consequently activates the patient’s immune system against cancer [103,104]. In a study of 437 ovarian and endometrial solid tumors, PD-1 expression was found in 80% to 90% of cases [105]. An example is Pembrolizumab, which is used as a promising therapy in carcinomas showing loss of MMR proteins such as melanoma and endometrial cancer [106,107,108]. Loss of function of the phosphatase tumor suppressor PTEN, which blocks the PI3K/AKT/mTOR pathway, is another area of research. The use of Temsirolimus, an mTOR inhibitor, was studied in a phase II study of 62 patients with recurrent metastatic endometrial cancer and demonstrated a remarkable response in patients who had not yet received any chemotherapy, regardless of PTEN status [109]. Unfortunately, in another phase II study of 42 patients with platinum-resistant ovarian cancer and advanced endometrial cancer, Temsirolimus treatment failed and the study was suspended [110]. Furthermore, patients with high microsatellite instability (MSI-high) also have a better response to immunotherapy. This is probably due to the fact that the excessive mutational load leads to an elevated expression of neo-antigens by each TILs-recalling cell, resulting in a response to immunotherapeutic drugs [111]. In May 2017, the US Food and Drugs Administration (FDA) accelerated the use of Pembrolizumab in patients with MSI-high or MMR protein loss in solid tumors. This was the first time the FDA has approved a treatment for patients with a specific molecular signature and not based on the location of the primary tumor. This decision was also supported by clinical studies in other cancer histotypes (NSCLC, melanoma, colon), which demonstrated that patients with high MSI molecular labeling or MMR protein deficiency had a very marked improvement in response outcomes [112,113,114,115]. These studies have been foundational and are leading to the validation of immunotherapy in endometrial cancer [116]. In 2017, the phase 1b study KEYNOTE-028 on the effect of Pembrolizumab on advanced or metastatic CE with PDL-1 positivity already treated with standard therapy showed a partial response in three patients, of which one had a mutation in POLE. The cumulative response was 13% with a six-month PFS of 19% and overall survival of 68.8%. Only mild adverse effects were found in 54.2% of patients [117]. Furthermore, in a recent phase Ib/II study, the combination of Pembrolizumab and the TKI Lenvatinib was tested in 23 patients with progressive metastatic EC, after standard chemotherapy. This study saw a 48% cumulative response with mild adverse effects [118]. Finally, there are numerous active clinical studies in the field of immunotherapy from which we expect promising answers in such a way as to be able to identify specific cohorts of patients also on the basis of molecular characteristics and genetic signatures.

## 7. Fertility-Sparing Options

Fertility-sparing treatments should be proposed to patients affected by AH/EIN/G1 with no myometrial invasion and who wish to have children. Continuous megestrol acetate, medroxyprogesterone, or levonorgestrel-IUD should be the medications of choice [119,120,121]. The NCCN defined patient selection criteria for fertility preservation: affected by well-differentiated endometrioid adenocarcinoma limited to the endometrium [122] on MRI/TVUS with the absence of suspicious/metastatic disease and no contraindications to progestin therapy and pregnancy with a close follow up with endometrial sampling every 3/6 months. We summarize the NCCN and ESGO guidelines for fertility-sparing management algorithm in Figure 3.

Hystopathology is the core strategy for stratification of the risk and the choice for fertility-sparing options for EC. The most common classification that may be useful to differentiate prognostic groups of EC implies the use of three IHC markers—TPp53, MSH6, PMS2—and the molecular test of POLE. In terms of prognostic values, EC should only be classified as POLE-mutated, which implies a low risk [123]. Some advance-stage ECs with POLEmut have excellent prognosis, while p53-abnormal (p53abn) tumors have poor prognosis [47,124]. MMRd or non-specific molecular profiles (NSMPs) have intermediate prognosis. New biomarkers, such as L1CAM or CTNNB1 mutation, may be useful to stratify low-grade endometrioid carcinomas [63]. An indicator of good response to EC is Dusp6, implicated in the MAP-Kinase cellular pathway, whereas MMR deficiency, mutations of PTEN, and overexpression of beta-catenin are indicators of therapy failure. Mutations of p53 and CTNNB1 are risk factors for recurrence. More studies are needed to assess the prognostic potential of new genes, such as KIF2C, CDK1, TPX2, and UBE2C [125]. Staging TAH/BSO is recommended after childbearing is completed or in case of progression. The reported complete response rate varies closely, depending on the stage and grade of EC. A durable CR occurs in about 50% of patients [119]. There is no uniform recommendation about progestin therapy timing, but maintenance treatment seems to lower the recurrence rate [126]. During follow up of patients who choose fertility-sparing therapy, clinicians should check symptom-oriented anamnesis and perform a complete clinical gynecological examination with a speculum, office echography, and rectovaginal palpation every 3/6 months during the first 3 years followed by every 6 months during the following 2 years. Cappelletti et al. performed a metanalysis of 42 studies that included 826 women about the possibility of achieving a pregnancy for EC patients treated with fertility-sparing progestin therapy (FSPT). In that study, the rate of live birth after FSPT was 20.5%. A complete response to fertility-sparing treatment was reported in 79.7% of EC patients, while the response rate of patients treated with FSPT was 79.9%. Recurrence was diagnosed in 35.3 per cent of a cohort of women with a previous complete response, and only one woman died during the follow-up. 

### Fertility-Sparing Treatment Outcomes

Of all women diagnosed with endometrial cancer, 6.5 percent are younger than 45 years old, and about 70 percent of them have not yet realized their wish to have children [127,128]. This new sociological milieu is responsible for the emergence of fertility-sparing treatments (FST) [129]. Young women should be informed by oncogynecologists about the strategies to improve fertility outcomes [130,131]. In a recent review [129] (*n* = 812), a complete or partial response to FST was found in about 83% of patients, and only a small percentage of patients were refractory to it. In 25.3% of cases a relapse occurred. Hormonal therapy plus hysteroscopy increased the pregnancy rate in EC-affected women to approximately 70% [132]. Some medications have been proposed for fertility-preservation strategies: Gonadotropin-RH agonists and aromatase inhibitors have had good outcomes in young and high-body-weight EC patients wishing to preserve their fertility [133,134,135]. In Table 3 we reported the results form Cappelletti et al. [136] about the mode of conception and pregnancy outcome after conservative management in early stage EC. Sub-fertility and worse obstetrical outcomes are multifactorial conditions linked to pre-existing metabolic disorders such as obesity, polycystic ovarian syndrome, insulin resistance, and a history of repeated curettage [134,135].

The literature reports a high rate of complications in women who reach pregnancy by medically assisted reproductive techniques. The most common obstetric complications are gestational diabetes, hemolysis and HELLP syndrome, and hypertension. These patients should be followed up in a highly specialized hospital in which specialists are aware of obstetric complications. It could be challenging to consider emerging risk factors to tailor FST to young oncological patients. In Table 4 we reported the general characteristics and rate of response to FST for EC in the main studies in literature.

## 8. New Prospectives in Non-Invasive Diagnostic and Prognostic Biomarkers for EC: The Proteomic Landscape

There is an emergent need for clinical noninvasive biomarkers for EC that could triage a selected cohort of high-risk patients to more invasive tests. The emerging relevance of translational sciences—transcriptomic, proteomic, and metabolomic—draws attention to the cellular, subcellular, and intercellular environments which are significant for tumorigenesis and tumor progression [168,169,170]. At present, these omics sciences are of less importance in the field of EC, and there is a need for further studies to assess their clinical significance in assessing tumor microenvironment and fingerprint [171,172], in addition to adapting the precise therapeutic strategy. Some interesting biomarkers for EC diagnostics (CA125, CA 19-9, HE4) and prognostics (L1CAM, COX2, SURVIVIN, CERB2) have been proposed. Unfortunately, none of those are currently used in clinical practice due to their lack of sensibility and specificity [173]. The most crucial blood biomarker candidates for EC detection are prolactin, he4, cancer antigens, YKL-40, and adiponectin [168]. Prolactin has over 98% sensitivity and specificity for EC, but is also elevated in the pancreas, ovary, and lung cancers [174]. Human Epididymis Protein 4 has 90% specificity for detecting EC [175], but is also overexpressed in other tumors. However, these proteins are useful in association with other biomarkers: the association of HE4 with SERUM AMILOID A has 73 specificity and 64% specificity in EC diagnostics, respectively, and HE4 with CA125 has over 60% specificity and 90% specificity in EC diagnostics, respectively [176,177,178,179]. Antigens CA125, CA72.4, and CA15-3 have a sub-optimal diagnostic accuracy because they are not specific [180]. Other blood markers include Human chitinase-3-like protein1 (YKL-40) [181] and Adiponectin/Leptin. The most helpful protein candidates for EC detection in endometrial uterine lavage specimens are chaperonin 10, Pyruvate kinase, and serpina1. Chaperonin 10, a chaperone involved in protein folding, is upregulated in EC tissues but is elevated in many other conditions. There is evidence that Pyruvate Kinase, a protein involved in the glycolytic pathway, is upregulated in EC tissues. Like Chaperone 10, it is not specific because it is related to other malignant and physiologic metabolic conditions. There is limited evidence regarding Serpina 1, a serine protease inhibitor downregulated in EC tissues. However, combining the three biomarkers, CPN 10, PK, and SERPINA1, showed sensitivity, specificity, and positive predictive values of over 95% for EC diagnostic. In the field of prognostic markers, there is a growing attention to the tumor heterogenicity, i.e., the co-existence of distinct subpopulations of cancer cells with different metabolic pathways and signaling profiles in the same tissue. This field of study promises to explain why tumors with the same histology may differ in their response to target therapy [182,183,184,185,186]. Annexin 2, a phospholipid-binding protein that plays a crucial role in cell growth and signal pathway transduction, is one of the candidates for predicting the recurrence of EC in in vitro, experiments but there is a need for clinical studies to assess the use of this biomarker in clinical settings [187]. One field of interest in the assessment of therapeutic strategies for endometrial cancer is the prediction of response to adjuvant treatment. There are some genomic markers proposed by TCGA that can divide cancers by risk profile. The change in proteome profile during EC adjuvant therapy needs further studies to evaluate possible markers of response or recurrence to selected therapies and to propose fertility-sparing options to patients who will respond to conservative treatment [188].

## 9. Conclusions and Perspective

There have been recent advancements in the area of fertility-sparing treatments for early stage endometrial cancer, but more studies are needed to fully understand their safety and efficacy. Factors such as the patient’s age, BMI, markers of ovarian reserve such as anti-Müllerian hormone and antral follicle count, and the stage and type of cancer can all impact the decision-making process. It is important for patients to have an open and honest discussion with their doctor and to be supported by a multidisciplinary team of medical specialists. The 2022 ESGO/ESHRE/ESGE Guidelines recommend referring patients with a pregnancy wish to a specialized center with a multidisciplinary team of gynecologic oncologists, fertility specialists, pathologists, and radiologists to aid in the decision-making process. Histological, molecular, and clinical features guide EC treatments. However, advanced research in translational science on carcinogenesis is revolutionizing the standard therapy of most cancers. EC is not yet benefiting from tailored therapies compared to other malignancies. Systemic traditional CHT is the only treatment international guidelines recommend for advanced and recurrent ECs. Targeted therapies should be one of the solutions in research settings for treating ECs. In endometrioid EC, the loss of function of PTEN, the activating mutation of PIK3CA, and the mutation of ARID1A are some of the common mutations. Intriguingly, ARID1A is a regulator of DNA damage checkpoint, leading the way to the use of Poly (ADP-Ribose) Polymerase (PARP) inhibitor. Dysregulation of the HER2 molecule tyrosine kinase is another attractive druggable molecular target in clear cell EC. Molecular aberrations that lead to carcinogenesis are still under study. Serous ECs are primarily independent of estrogen and are TP53-mutated. Common mutations in carcinosarcoma imply TP53, PTEN, PIK3CA, and PIK3R1, and are potentially targetable with an immune checkpoint inhibitor. Mismatch Repair Deficiency with mutations of the MMR genes MLH1, MSH2, MSH6, PMS2, or EPCAM leads to the accumulation of neoantigen loads, with promising outcomes with immune checkpoint inhibitors. Low-grade metastatic/recurrent endometrioid ECs with the expression of estrogen receptors have a reasonable response rate to hormonal therapy with drugs such as tamoxifen and megestrol/medroxyprogesterone acetate. Immunotherapies with checkpoint inhibitors could be effective in MSI-high/MMR-deficient or -high TILs. With the expression of PD-ligand1-2, activation of the PI3K/Akt/mTOR pathway leads to mTOR inhibitors that only have low activity. Additionally, overexpression of HER2, which is frequent in serous ECs, has a small clinical utility: treatment with trastuzumab lacks efficacy. ARID1A deficiency has a potential clinical utility for the use of PARP inhibitors. It is crucial for patients to be aware that pregnancy after endometrial cancer treatment may come with increased risks and to carefully consider the potential benefits and risks of fertility-sparing options. Achieving a pregnancy after conservative treatment for endometrial cancer could be followed by obstetrical complications. New predictive markers could assist reproduction specialists in helping these women conceive and predict and avoid complications.

## Figures and Tables

**Figure 1 ijms-24-09780-f001:**
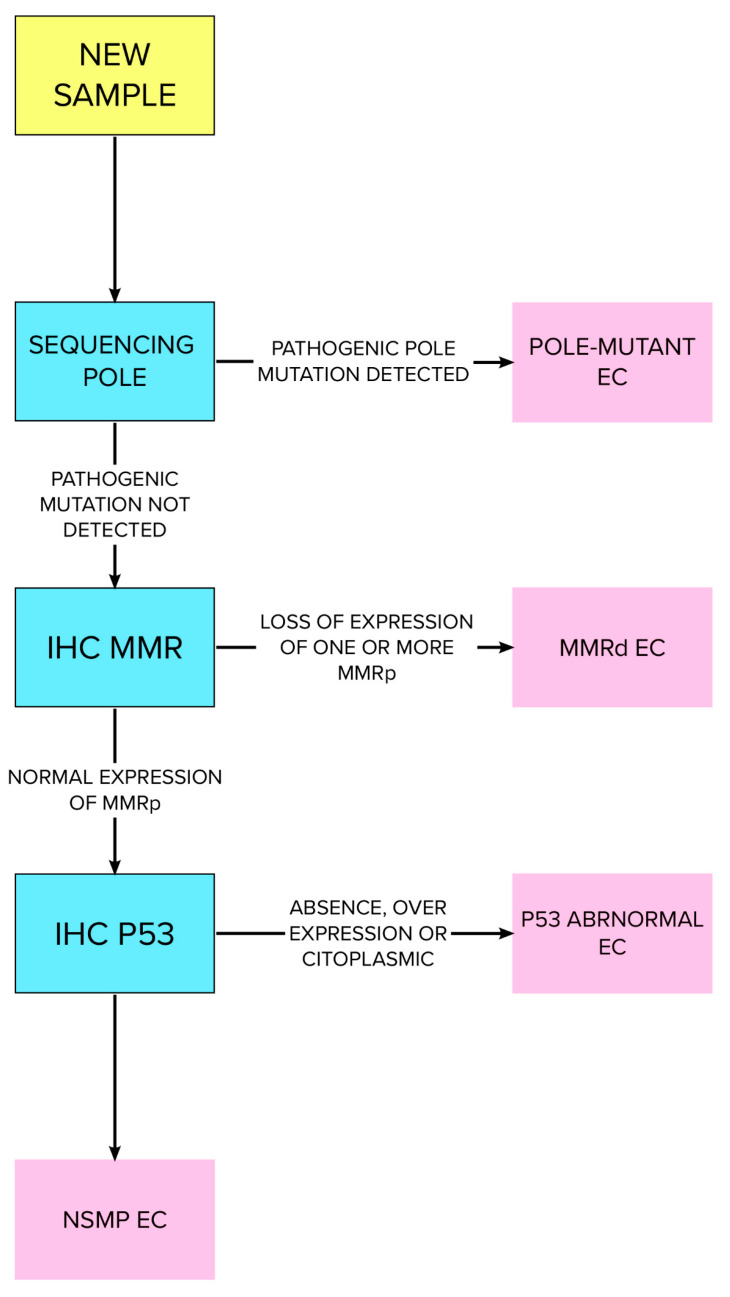
WHO-proposed algorithm for testing EC pathological specimens. Rearranged and modified from Crosbie et al. [55].

**Figure 2 ijms-24-09780-f002:**
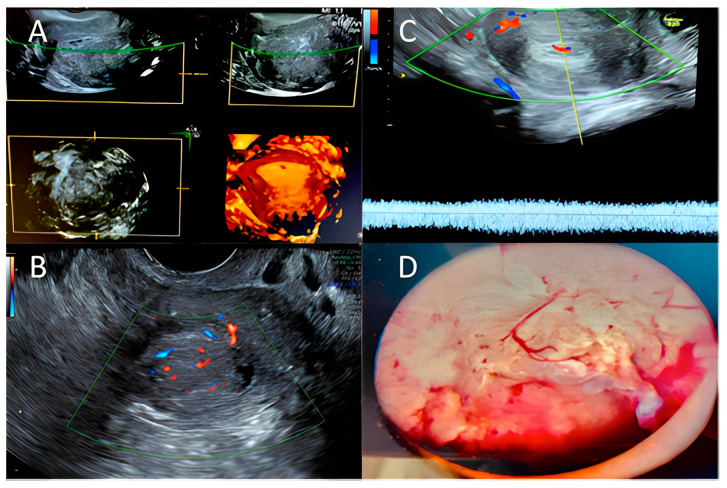
Hyperplasia/endometrial cancer: (**A**) 3D ultrasound exam; (**B**,**C**) Color–Doppler score examination—Courtesy of MD and VP; (**D**) Hysteroscopic typical pattern for endometrial hyperplasia/endometrial cancer. Courtesy of GRD.

**Figure 3 ijms-24-09780-f003:**
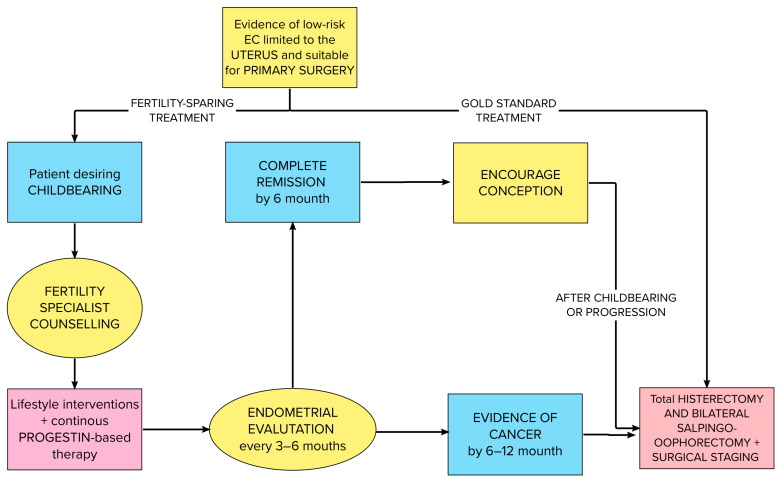
Integrated NCCN and ESGO guidelines flow chart for fertility-sparing management.

**Table 1 ijms-24-09780-t001:** ESGO–ESP–ESTRO prognostic groups of risk modified and rearranged from Crosbie et al. [55]. Non-E.C = Non E.C. carcinoma: serous, clear cell, carcinosarcoma, mixed, undifferentiated carcinoma. E.C. = E.C. carcinoma. LVSI = lymphovascular space invasion. NSMP = non-specific molecular pattern. St. = Stage.

	Molecular Classification				Molecular Classification
	Known				Unknown
	Pole-Mutant	Mmr-Deficient	NSMP	P53 Abnormal	
Low risk	St. I-II, no residual disease	St. IA, E.C Low-grade, negative/focal LVSI			St. IA, E.C., low-grade, with negative
					or focal LVSI
Intermediate risk		St. IB E.C Low-grade with negative/focal LVSI		St. IA without myometrial invasion	St. IB, E.C. low-grade, negative or focal LVSI
		St. IA E.C., high-grade with			St. IA, E.C. high-grade, negative or focal LVSI
		negative/focal LVSI			St. IA non-E.C. without myometrial invasion
		St. IA non-E.C., without myometrial invasion			
High-intermediate risk		St. I E.C., with substantial LVSI, regardless of grade or depth		St. I-IVA with myometrial invasion and no residual disease	St. I E.C. with substantial LVSI, regardless of grade or depth of invasion
		of invasion			St. IB, E.C. high-grade, regardless of LVSI
		St. IB high-grade E.C. with any LVSI			St. II E.C.
		St. II EC			
High risk		St. III-IVA E.C., with no residual disease		St. I–IVA, with	St. III–IVA E.C. with no residual disease
		St. I-IVA non E.C. with no myometrial invasion and no residual disease		myometrial invasion and no residual disease	St. I–IVA non-E.C with myometrial invasion and no residual disease
Advanced/metastatic	St. II-IVA with residual disease				St. II–IVA with residual disease
	St. IV B				St. IVB

**Table 2 ijms-24-09780-t002:** FIGO staging of uterine corpus carcinoma and carcinosarcoma, TNM, and AJCC, modified and integrated from Koska et al. [88,89].

TNM STAGE	FIGO STAGE	Short Definition	Details
T1	I	Confined to the uterine corpus	EC inside the uterus, and/or the cervical glands. Not in the cervical stroma. No nearby lymph nodes (N0). No metastasis (M0).
N0			
M0			
T1a	IA	Involves < 50% of the myometrium	EC is in the endometrium and may have grown <50% the myometrium (T1a). No nearby lymph nodes (N0). No metastasis (M0).
N0			
M0			
T1b	IB	Invasion ≥ 50% of the myometrium	EC has grown ≥50% of myometrium, NOT beyond the uterus (T1b).
N0			
M0			
T2	II	Invasion of the cervical stroma but no extension outside the uterus	EC spread from the uterus body and growing into the cervical stroma. Not spread outside the uterus (T2).
N0			
M0			
T3	III	Local and/or regional spread of the tumor	EC is outside the uterus, but NOT in rectum or urinary bladder (T3). No nearby lymph nodes (N0). No metastasis (M0)
N0			
M0			
T3a	IIIA	Invasion of uterine serosa, adnexa, or both (direct extension or metastasis)	EC outside the serosa of the uterus and/or to the adnexa (T3a).
N0			No nearby lymph nodes (N0). No metastasis (M0).
M0			
T3b	IIIB	Metastases or direct spread to the vagina and/or spread to the parametria	EC in the vagina or in the parametrium (T3b).
N0			No nearby lymph nodes (N0). No metastasis (M0).
M0			
	IIIC	Metastases in pelvic or para-aortic lymph nodes, or to both	
T1-T3	IIIC1	Metastases to pelvic lymph nodes	EC has extended to some nearby tissues, but NOT into the inside of the urinary bladder or rectum (T1 to T3).
N1, N1mi or N1a			Spread to pelvic lymph nodes (N1, N1mi, or N1a), but NOT to aorta lymph nodes or distant sites (M0).
M0			
T1-T3	IIIC2	Metastases to para-aortic lymph nodes, with or without metastases to pelvic lymph nodes	
N2, N2mi or N2a			Spread to lymph nodes around the aorta (para-aortic lymph nodes) (N2, N2mi, or N2a), but not to distant sites (M0).
M0			
	IV	Involvement of the bladder and/or intestinal mucosa and/or distant metastases	
T4	IVA	Invasion of the bladder, intestinal mucosa, or both	Extend to the inner lining of the rectum or urinary bladder (called the mucosa) (T4).
Any N			M0
M0			
Any T	IVB	Distant metastases, including metastases to the inguinal lymph nodes or intraperitoneal disease	Inguinal lymph nodes are positive, and/or distant metastases (lungs, liver, bones).
Any N			Any size (Any N).
M1			

**Table 3 ijms-24-09780-t003:** Mode of conception and pregnancy outcome after conservative management of early stage endometrial cancer. (Modified and rearranged from Cappelletti et al. [136]).

Conception	Percentage of Pregnancy
Spontaneous	27.6
Fertility treatment	55.2
**Outcome**	**Percentage**
Miscarriage	26.9
Ongoing pregnancy	3.5
Delivery and live birth	68.9
**Multiple birth**	
No	71.1
Yes, twin	5.6
Yes, 3	1.5

**Table 4 ijms-24-09780-t004:** General characteristics and rate of response of main studies on conservative treatment of EEC. (Modified from Cappelletti et al. [136]). (Ch: Cohort, Cs: case series, Os: oral, IU: intrauterine, Hr: Hysteroscopic resection).

Reference (First Author—Country)	Study Design	Subjects with EEC	Mean Age (Years)	Route of Progestin Administration	Additional Treatment	Subjects with Complete Response	Complete Response (%)	Mean Follow-Up (Months)
Andress et al. (2021)—de [137]	Ch	10	34	OS		5	50%	16.7
Ayhan et al. (2020)—tr [138]	Ch	30	32	OS AND/OR IU	Hr	22	73.3%	55.5
Cade et al. (2013)—Au [139]	Ch	10	32	OS AND/OR IU		10	100%	89.2
Casadio et al. (2020)—It [140]	Cs	36	33	OS	Hr	35	97.2%	30
Chen et al. (2016)—Cn [141]	Ch	37	32	OS		27	73%	54
Choi et al. (2013)—Kr [142]	Cs	11	31		Photodynamic therapy + IV photosensitizer	7	63.6%	82.7
Duska et al. (2001)—Us [143]	Ch	12	30	NR		10	83.3%	NR
Falcone et al. (2017)—It [144]	Ch	27	36	OS OR IU	Hr	26	96.3%	96
Giampaolino et al. (2019)—It [145]	Cs	14	35	IU	Hr	11	78.6%	NR
Kaku et al. (2001)—Jp [146]	Cs	10	30	OS		7	70%	33.6
Kim et al. (2013)—Kr [147]	Cs	16	34	OS AND IU		14	87.5%	31.1
Kudesia et al. (2014)—Us [148]	Ch	10	38	OS AND/OR IU		7	70%	21.3
Maggiore et al. (2019)—It [149]	Cs	16	33	IU		13	81.3%	85.3
Minaguchi et al. (2007)—Jp [150]	Ch	19	30	OS		15	78.9%	45.1
Minig et al. (2011)—It [151]	Ch	14	34	IU	GnRH agonists	8	57.1%	29
Niwa et al. (2005)—Jp [152]	Cs	10	30	OS		10	100%	52.2
Ohyagi-Hara et al. (2015)—Jp [153]	Cs	16	NR	OS		11	68.8%	NR
Ota et al. (2005)—Jp [154]	Cs	12	30	OS		5	41.7%	52.7
Park et al. (2013)—Kr [155]	Ch	177	NR	OS		14	79.7%	NR
Pashov et al. (2012)—Ru [156]	Cs	11	30	IU	GnRH agonists	11	100%	44.4
Perri et al. (2011)—Il [157]	Ch	25	NR	OS OR IM		22	88%	NR
Raffone et al. (2021)—It [158]	Ch	6	35	IU	Hr	2	33.3%	NR
Shan et al. (2013)—Cn [159]	Ch	14	30	OS		11	78.6%	34.7
Shirali et al. (2012)—Ir [160]	Ch	16	33	OS		10	62.5%	NR
Ushijima et al. (2007)—Jp [161]	Ch	28	31	OS		14	50%	47.9
Wang et al. (2014)—Tw [162]	Ch	37	32	OS	Hr	30	81.1%	78.6
Wang et al. (2017)—Cn [163]	Cs	11	27	OS OR IM	Hr	9	81.8%	82.3
Yamagami et al. (2018)—Jp [164]	Ch	97	35	OS		88	90.7%	71.3
Yamazawa et al. (2007)—Jp [165]	Ch	9	36	OS		7	77.8%	38.9
Yu et al. (2009)—Cn [166]	Cs	8	25	OS OR IM		5	62.5%	31.8
Zhou et al. (2015)—Cn [167]	Ch	19	30	OS		15	78.9%	32.5

## Data Availability

Not applicable.

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
