# Peer review of "Upgrading Treatment and Molecular Diagnosis in Endometrial Cancer—Driving New Tools for Endometrial Preservation?"

_ijms, 2023, doi:10.3390/ijms24119780_

Round 1

Reviewer 1 Report

I read with great interest the manuscript " ,  Upgrading in Treatment and Molecular Diagnosis in Endometrial cancer driving new tools in endometrial preservation? , submitted to IJMS . The topic is attractive for several reasons: recent improved knowledge about molecular portraits of endometrial cancer , new possibilities for tailored strategies of endometrial preservations, innovative therapeutical  approaches with targeted tagents  and immunotherapeutic agents, the current and future strategies finalized to fertility preservation in young patients affected with endometrial cancer  .The abstract is clear, exhaustive and well related with the the topics . The different manuscript sections are complete ,propositive , documented ( epidemiology, risk factors, diagnostic work up, novel molecular classification, surgical updating, staging, traditional and novel therapeutic challenges, fertility preservations strategies ). The bibliography list is complete and updated.  The tables, figures and images are clear , detailed and easily understandable. 

Because the text is very rich , it is important to check again and carefully the english language in order to avoid errors and misinterpretations of the data and concepts expressed.

Moreover I suggest to introduce in the bibliographic list the following papers in order to improve the quality of this very interesting manuscript:

Makker V, MacKay H, Ray-Coquard I et al . Endometrial cancer.Nat Rev Dis Primers. 2021 Dec 9;7(1):88. doi: 10.1038/s41572-021-00324-8.

Lee AJ, Yang EJ, Kim NK et al.Fertility-sparing hormonal treatment in patients with stage I endometrial cancer of grade 2 without myometrial invasion and grade 1-2 with superficial myometrial invasion: Gynecologic Oncology Research Investigators coLLaborAtion study (GORILLA-2001).

Gynecol Oncol. 2023 May 10;174:106-113. doi: 10.1016/j.ygyno.2023.04.027.

Musacchio L, Boccia SM, Caruso G et al.Immune Checkpoint Inhibitors: A Promising Choice for Endometrial Cancer Patients? J Clin Med. 2020 Jun 3;9(6):1721. doi: 10.3390/jcm9061721.

Mustea A, Ralser DJ, Egger E et al.Determination of the Cancer Genome Atlas (TCGA) Endometrial Cancer Molecular Subtypes Using the Variant Interpretation and Clinical Decision Support Software MH Guide.Cancers (Basel). 2023 Mar 30;15(7):2053. doi: 10.3390/cancers15072053.

Tomao F, Peccatori F, Del Pup L et al.  Special issues in fertility preservation for gynecologic malignancies.Crit Rev Oncol Hematol. 2016 Jan;97:206-19. doi: 10.1016/j.critrevonc.2015.08.024.

The english language is acceptable , but because the text is very rich , it is important to check again and carefully the text in order to avoid errors and misinterpretations of the data and concepts expressed.

Author Response

Dear Reviewer,

Once again we want to thank you for your valuable advice. Attached you will find our report on the changes made to the manuscript.

Reviewer 2 Report

For more than a decade of development, molecular diagnosis and molecular prognosis become more important in cancer therapeutics.  This manuscript from Dellino et al provided sufficient summary and discussion for the progressive innovation in the management of endometrial cancer.

1. The authors summarised the conventional diagnosis and classification and associated risk factors in Paragraph 1 and 2.

2.  The authors summarised the ongoing risk assessment from phenotype-genotype prediction models according to the WHO consideration.

3.    The authors summarised the implementation of EC staging by advanced imaging systems in Paragraph 4 and 5. Moreover, the NCCN guideline and some recent advanced considerations regarding the immunotherapy in EC were also included.

Indeed, since endometrial preservation for EC patient was the main content of the manuscript, I would like to suggest the authors that enhance the discussion of this points views with the summarised studies in Paragraph 7 Fertility sparing options:

1.  Table 4 has listed out the General characteristics and rate of response of main studies on conservative treatment of EEC.  Are there sufficient data to run meta-analysis to evaluate the validity of these studies?

2.   Would the authors suggest an enhanced guideline (e.g. graphical flowchart based on Figure 1 WHO suggestion) for the treatment plan of EC patients who require endometrial preservation?

3.   Since the authors has considered the use of TCGA platform, is there possible to discuss the use of or cooperative research of CPTAC (NIH OCCPR) proteomics platform (https://proteomics.cancer.gov/programs/cptac) to study discrepancy of protein profile of endometrial layers of the EC patient in pre-treatment (before the adjuvant therapies) and post-treatment conditions, and the association of pregnancy outcomes?

Author Response

(The authors gave the same response as above.)
